# Assessment of Climate Change and Land Use Effects on Water Lily (*Nymphaea* L.) Habitat Suitability in South America

John M. Nzei [1,2,3], Boniface K. Ngarega [4], Virginia M. Mwanzia [5], Joseph K. Kurauka [6], Qing-Feng Wang [1,2,7], Jin-Ming Chen [1,2], Zhi-Zhong Li [1,2,*] and Cheng Pan [1,*]

1 Wuhan Botanical Garden, Chinese Academy of Sciences, Wuhan 430074, China
2 Center of Conservation Biology, Core Botanical Gardens, Chinese Academy of Sciences, Wuhan 430074, China
3 University of Chinese Academy of Sciences, Beijing 100049, China
4 Centre for Integrative Conservation, Xishuangbanna Tropical Botanical Garden, Chinese Academy of Sciences, Menglun 666303, China
5 School of Agriculture and Technical Studies, Lukenya University, Mtito Andei P.O. Box 90-90128, Kenya
6 School of Environmental Studies, Kenyatta University, Nairobi P.O. Box 43844-00100, Kenya
7 Sino-Africa Joint Research Center, Chinese Academy of Sciences, Wuhan 430074, China
* Correspondence: lizhizhong@wbgcas.cn (Z.-Z.L.); pancheng@wbgcas.cn (C.P.)

**Abstract:** Many aquatic species have restricted dispersal capabilities, making them the most vulnerable organisms to climate change and land use change patterns. These factors deplete *Nymphaea* species' suitable habitats, threatening their populations and survival. In addition, the species are poorly documented, which may indicate how scarce they are or will become. Members of *Nymphaea* are ecologically important as well as having cultural and economic value, making them of conservation interest. Therefore, using the maximum entropy (MaxEnt) approach, climatic variables, land use, and presence points were modeled for seven *Nymphaea* species in South America, using three general circulation models (CCSM4, HADGEM2-AO, and MIROC5) and in two representative concentration pathways (RCPs 4.5 and 8.5) and two scenarios (2050 and 2070). Our results indicated that mean diurnal range (bio2), precipitation of the wettest month (bio13), temperature seasonality (bio15), and land use (dom_lu) were the main influencing factors. For all species, suitable areas were concentrated east of Brazil, and they were variable in northern parts of the continent. Besides, inconsistent expansion and contraction of suitable habitats were noticed among the species. For example, *N. amazonum*, *N. rudgeana*, and *N. lasiophylla* future habitat expansions declined and habitat contraction increased, while for *N. ampla* and *N. jamesoniana,* both future habitat expansion and contraction increased, and for *N. pulchella* and *N. rudgeana* it varied in the RCPs. Moreover, the largest projected suitable habitats were projected outside protected areas, characterized by high human impacts, despite our analysis indicating no significant change between protected and unprotected areas in suitable habitat change. Finally, understanding how climate change and land use affect species distribution is critical to developing conservation measures for aquatic species.

**Keywords:** climate change; distribution; habitat suitability; *Nymphaea*; land use; conservation

## 1. Introduction

Temperature and precipitation fluctuations have been reported in South America over the last decade [1]. For example, in Brazil, the temperature has increased by approximately 0.5 °C [2], while the mean temperature variability across the continent varied between 0.2 and 0.8 °C, with a projected increase of 1 to 4 °C by the end of the century [3]. The average precipitation is projected to increase in the southeast and decrease in other areas, especially between latitudes 5° and 20° south [1]. The effect of climate change on the continent is also seen in the rapid melting of glaciers, which is associated with changes in temperature and humidity [4,5]. As global warming continues to rise, species respond to climate change, causing increased shifts and redistributions in search of suitable habitats.

Ultimately, this has an impact on conservation and management plans in maintaining biodiversity [6].

As the earth becomes much warmer, various land regions become drier, such as parts of the Amazon Forest in Brazil which are being replaced by non-forest environments [7]. Global warming is also linked to the increased savannah land in the Amazon and semi-arid areas northeast of Brazil, which are slowly transforming to deserts [8]. Besides climate warming, the increased human activities pose negative consequences for many parts of the region, such as the Caatinga biome east of Brazil [8]. As climate change affects species community assemblages, the direct effect of human activity in the ecosystem is loss of habitat, which is the primary loss of biodiversity [9]. The unprecedented population growth, urbanization, and need for sustainable food production are among the major humanized factors leading to ecosystem exploitation and loss of biodiversity. These activities also create large disruption to stream flow and wetlands, as well as influencing the hydrological events that lead to floods and droughts, which further threaten extinction of wetland ecosystems and their biodiversity.

The distribution of *Nymphaea* species is considered widespread worldwide [10], however in South America the distribution is poorly explored and documented. Their distribution is particularly prone to climate change and human activities, more so from the characteristic assemblages of the presence data. Most occurrence points were obtained from Brazil compared to the other states in the continent, which might signify inconsistent availability of the species or sampling in the most accessible areas [11]. Much of their restricted ranges threaten the species' habitat suitability with a risk of decline or extinction. This may signify the loss of wild populations that play a key role not only scientifically or culturally but also ecologically in providing food, habitat, water sediments, and turbidity management, and as an indicator for a healthy wetland ecosystem [12,13]. Besides, other studies have indicated climate change to be of concern in aquatic species' distribution [14–16]. Although human influence was not included in those studies, its impact, combined with climate change, poses an unlimited threat to biodiversity and habitat loss for the species [17]. For example, it is approximated that 15% (560,000 km$^2$) of the forested area in Brazil has so far been lost to ranching and agriculture [18,19]. Considering the sensitivity of the species habitat environment and the species distribution data deficit, their vulnerability to climate change and human influence make them a great choice for this study.

The ecological niche models (ENMs) approach has been employed as a valuable tool for assessing the species habitat suitability [20], enabling insights for conservation measures. These models have been widely used to estimate potentially suitable habitats for a variety of species in temporal and spatial ranges using species occurrence records and environmental variables, in addition to species range shifts, habitat quality, habitat requirements, and to identify potential distribution regions for species. However, maximum entropy (MaxEnt) is much preferred for its ability to accommodate presence only data [21], it performs well and can work with small sample sets [22–24], and avoids commission errors when projecting species distribution [25], thus making it suitable in the assessment of aquatic organism-suitable habitats. It also performs well on both large and narrow geographical distribution scales, such as in the distribution of *Ottelia* and water lily species across Africa and Australia, respectively [14–16].

Using the MaxEnt modeling approach, we assessed the habitat distribution for seven *Nymphaea* species in South America. Our goals were to: (i) model and predict the current distributional range for *Nymphaea* species in South America, (ii) identify the environmental variables shaping the habitat and distribution of *Nymphaea* species, (iii) predict the future habitat suitability of the species, (iv) evaluate human effects on the distribution of the water lilies, and (v) predict the percentage threat of the water lilies' suitable habitats by evaluating the habitat suitability under protected areas (PAs) and unprotected areas (un-PAs).

## 2. Materials and Methods

### 2.1. Species Occurrence Data

The species occurrence data for *Nymphaea* were obtained from the Global Biodiversity Information Facility using rgbif package in R (GBIF; https://www.gbif.org/, retrieved on 6 February 2022) and from [26,27]. The downloaded distribution localities were manually filtered to remove duplicate samples and coordinates with ambiguous geographic localities. Google Earth (https://earth.google.com, accessed on 7 February 2022) was used to examine the precision of the coordinates, and those with apparent errors in their geographic coordinates were eliminated. The remaining localities for each species were then rarefied to a spatial distance of 5 km between the points to reduce spatial autocorrelation. This analysis was implemented in R software with the package spThin [28]. The remaining points included *N. amazonum* Mart. and Zucc. (93), *N. ampla* (Salib.) DC. (32), *N. jamesoniana* Planch (23), *N. lasiophylla* Mart. and Zucc. (47), *N. lingulata* Wiersma (47), *N. pulchella* DC (138), and *N. rudgeana* G. Mey. (62) (Figure 1; Table S1).

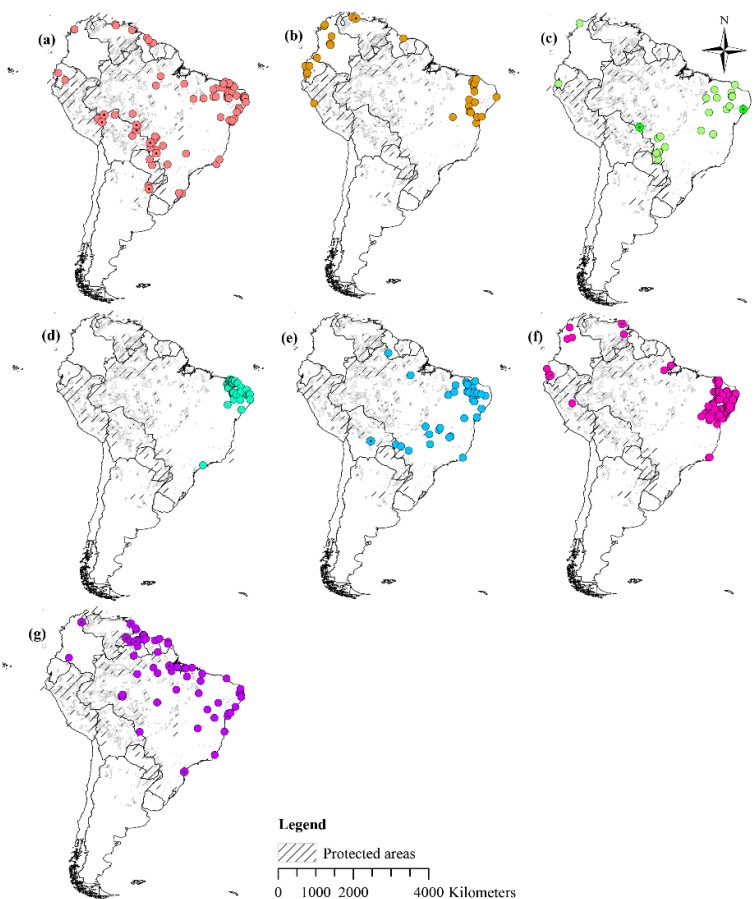

**Figure 1.** Occurrence points for the seven water lily (*Nymphaea*) species in South America: (**a**) *N. amazonum*, (**b**) *N. ampla*, (**c**) *N. jamesoniana*, (**d**) *N. lasiophylla*, (**e**) *N. lingulata*, (**f**) *N. pulchella*, and (**g**) *N. rudgeana*. The points marked with a black dot inside indicate occurrence points within protected areas.

### 2.2. Environmental Data

Nineteen bioclimatic variables spanning 1950–2000 were downloaded from World-Clim v1.4 (http://worldclim.org/version2, accessed on 10 April 2021 [29]) at a 2.5 arc-min spatial resolution. These variables define yearly climatic temperature and precipitation trends, as well as seasonality and extreme factors that could exert physiological limits on organisms and influence their geographic distribution. In addition, the land use variable was obtained from the Food and Agricultural Organization (https://www.fao.org/, accessed

on: 16 June 2022) at a spatial resolution of 30 arc sec (1 km) and resampled to match the bio-climatic variables' resolution in ArcGis 10.8 [30]. The variables were then masked to the M area of the BAM diagram [31] using South America's freshwater ecoregions as the species' accessible regions [32]. Bioclimatic variables bio8, bio9, bio18, and bio19 were omitted from further analysis as they are known to contain some artifacts and express unrealistic climatic changes between adjacent pixels, although the discontinuities in the variables may not have significant biological meaning [33,34]. Then, we performed the variance inflation factor (VIF: r = 0.7) in the usdm package in the R program to eliminate correlated variables [35] in 16 variables (Table S2). Three general climate models (GCMs): Community Climate System Model version 4 (CCSM4) [36], Hadley Centre Global Environment Model version AO (HADGEM2-AO), and the Model for Interdisciplinary Research on Climate (MIROC5), were selected for the future habitat suitability prediction as they are less biased and have been shown to produce good modeling results [16,22,37]. We selected two representative concentration pathways (RCPs), RCP 4.5 and RCP 8.5, in two scenarios (2050 and 2070) to represent medium- and high-emission scenarios consistent with the IPCC Fifth Assessment Report [3].

### 2.3. Model Parameterization and Calibration

The climatic niche of *Nymphaea* species was modeled using MaxEnt v.3.4.1. [38]. The model was built with default settings, except for 5000 iterations, 10 replications, cross-validated run-type, and 75% and 25% of occurrence data for training and testing of the model, respectively. To control over-fitting, regularization multiplier (rm) and feature classes were selected using ENMeval in R program in five feature classes (L, LQ, LQH, LQHP, LQHPT, being linear (L), quadratic (Q), hinge (H), product (P), and threshold (T)) at rm values of 0.5 to 4.0 at a 0.5 increment [39]. To evaluate the predictive performance of the models, we utilized the area under the curve of the receiver operating characteristic curve (ROC) [40].

### 2.4. Predicting Current and Future Range Shifts

Using a threshold criterion of maximum training sensitivity plus specificity (MTSS), the average 'Logistic' outputs were converted to binary maps depicting climatic suitable and unsuitable areas. In this step, the continuous suitability outputs were changed to binary maps so that over 95% of the training occurrence data fell inside the suitable range. Finally, using the binary maps, current and future habitat suitability change were assessed using the SDM-Toolbox extension in ArcGis 10.8 [30,40,41]. The generated maps show habitat suitability changes in (i) stable, (ii) expansion, (iii) contraction, and (iv) unsuitable areas.

### 2.5. Species Conservation/Threat Area

To assess the current and future possible habitat threat to the species, a map of PAs was obtained from world protected planet (available at: https://www.protectedplanet. net/search?q=natura+2000, accessed on: 24 May 2022) and overlaid with the species' occurrences to assess the percentage of populations inside PAs, the current distribution, and the future projection percentage change of suitable habitats inside and outside PAs. Further, the area of land use in both PAs and un-PAs was used to assess the likely influence of human activities in the species' area of distribution.

## 3. Results

### 3.1. Variable Selection and Model Performance

After the correlation analyses, mean diurnal range (bio2), temperature seasonality (bio4), the maximum temperature of the wettest month (bio5), precipitation of the wettest month (bio13), precipitation seasonality (bio15), and Land use (dom_lu) were used in the model building. The selected variables had VIF values of less than three (Table 1). The ENMeval analysis predicted three feature classes (LQ, LQH, and LQHP) and rm values of

2–4 as the best parameters for the models (Table 2). Besides, the AUC values were above 0.8 in all projection scenarios (Table 3), demonstrating high model accuracy. This suggests that the produced models outperformed a random model (AUC = 0.5), indicating that the suitability of the models in the forecast for the distributions of these seven *Nymphaea* species was reliable. In addition, the mean training and testing AUC values had no significant difference and the standard deviation values were close to the probability distribution (Table S3).

**Table 1.** Climatic variables retained after correlation analysis using variance inflation factors (VIF).

| Variable No. | Bioclimatic Variable | Code | VIF |
|---|---|---|---|
| 1 | Mean diurnal range (mean of monthly (max temp – min temp)) | bio2 | 1.7436 |
| 2 | Temperature seasonality (standard deviation × 100) | bio4 | 1.9289 |
| 3 | Maximum temperature of the warmest month | bio5 | 1.1728 |
| 4 | Precipitation of the wettest month | bio13 | 1.8567 |
| 5 | Precipitation seasonality (coefficient of variation) | bio15 | 1.4996 |
| 6 | Land use cover | dom_lu | 1.2866 |

**Table 2.** Feature classes selected for the modeling of the *Nymphaea* species in South America. The abbreviations represent linear (L), quadratic (Q), hinge (H), and product (P) at varying regularization multiplier (rm) values.

| Species | Features Class | rm Value | Current Habitat Suitability (km²) |
|---|---|---|---|
| *N. amazonum* | LQHP | 3.5 | 2,339,884 |
| *N. ampla* | LQ | 2.5 | 1,558,143 |
| *N. jamesoniana* | LQH | 4 | 2,790,903 |
| *N. lasiophyla* | LQH | 2 | 862,217 |
| *N. lingulata* | LQH | 4 | 1,341,940 |
| *N. pulchella* | LQH | 4 | 907,696.1 |
| *N. rudgeana* | LQH | 3.5 | 2,657,233 |

**Table 3.** The area under the curve (AUC) and standard deviation values for the seven *Nymphaea* species distribution models in two representative concentration pathways (RCPs 4.5 and 8.5) and two scenarios (2050 and 2070).

| Species | Current | RCP 4.5 | | RCP 8.5 | |
|---|---|---|---|---|---|
| | | 2050 | 2070 | 2050 | 2070 |
| *N. amazonum* | $0.847 \pm 0.075$ | $0.808 \pm 0.116$ | $0.803 \pm 0.118$ | $0.815 \pm 0.115$ | $0.811 \pm 0.111$ |
| *N. ampla* | $0.878 \pm 0.081$ | $0.887 \pm 0.089$ | $0.889 \pm 0.090$ | $0.891 \pm 0.084$ | $0.879 \pm 0.085$ |
| *N. jamesoniana* | $0.842 \pm 0.083$ | $0.859 \pm 0.086$ | $0.848 \pm 0.096$ | $0.848 \pm 0.090$ | $0.848 \pm 0.094$ |
| *N. lasiophylla* | $0.963 \pm 0.036$ | $0.954 \pm 0.045$ | $0.951 \pm 0.049$ | $0.953 \pm 0.048$ | $0.958 \pm 0.035$ |
| *N. lingulata* | $0.902 \pm 0.036$ | $0.890 \pm 0.074$ | $0.889 \pm 0.075$ | $0.891 \pm 0.078$ | $0.882 \pm 0.087$ |
| *N. pulchella* | $0.927 \pm 0.025$ | $0.936 \pm 0.022$ | $0.935 \pm 0.021$ | $0.938 \pm 0.021$ | $0.933 \pm 0.023$ |
| *N. rudgeana* | $0.814 \pm 0.086$ | $0.794 \pm 0.083$ | $0.800 \pm 0.089$ | $0.796 \pm 0.089$ | $0.806 \pm 0.090$ |

### 3.2. Contribution of Variables

The percentage contributions of the bioclimatic variables to the final *Nymphaea* models are shown in Table 4. The best variables explaining the distribution of all *Nymphaea* species in all scenarios were bio2, bio13, dom_lu, and bio4. Bio2 had a limited contribution for *N. jamesoniana*. Bio4 indicated a greater contribution in the current projection compared to future, except for *N. rudgeana* and *N. lasiophyla*. Bio5 was the less contributing variable among the species in all scenarios except for *N. amazonum* and *N. jamesoniana*. Bio13 contributed greatly among the species in all scenarios except for *N. rudgeana*. Bio15 had much influence on the distribution of *N. lasiophylla*, *N. lingulata*, and *N. rudgeana*. Finally, dom_lu is of influence to all species' suitable habitats.

**Table 4.** The mean relative contribution for the six bioclimatic variables used for the habitat suitability modeling of the water lily species in South America in two representative concentration pathways (RCPs 4.5 and 8.5) and two scenarios (2050 and 2070).

| Species | Variable | Current | RCP 4.5 | | RCP 8.5 | |
|---|---|---|---|---|---|---|
| | | | 2050 | 2070 | 2050 | 2070 |
| *N. amazonum* | bio2 | **32.6** | **28.3** | **29.8** | **32.5** | **30.0** |
| | bio4 | 17.6 | 3.3 | 3.3 | 3.4 | 2.7 |
| | bio5 | 16.1 | 15.3 | 11.6 | 12.2 | 12.0 |
| | bio13 | 9.5 | **19.5** | **18.8** | **20.4** | **20.6** |
| | bio15 | 4.3 | 9.8 | 12.7 | 10.4 | 11.3 |
| | dom_lu | **20.0** | **23.7** | **23.9** | **21.2** | **23.3** |
| *N. ampla* | bio2 | **23.5** | **35.3** | **35.7** | **40.2** | **37.5** |
| | bio4 | 17.8 | 13.5 | 13.8 | 12.2 | 15.2 |
| | bio5 | 0.0 | 1.6 | 1.6 | 2.3 | 3.8 |
| | bio13 | **25.8** | **20.3** | **20.2** | **15.9** | **15.9** |
| | bio15 | 8.3 | 10.5 | 9.5 | 11.2 | 8.6 |
| | dom_lu | **24.6** | **18.9** | **19.2** | **18.1** | **19.2** |
| *N. jamesoniana* | bio2 | 0.0 | 0.3 | 0.4 | 0.2 | 0.6 |
| | bio4 | 18.6 | 6.4 | 6.8 | 3.9 | 4.5 |
| | bio5 | 29.1 | 16.6 | 14.3 | 12.6 | 8.9 |
| | bio13 | 2.8 | **20.9** | **17.0** | **22.7** | **21.1** |
| | bio15 | **19.3** | **21.3** | **23.5** | **23.2** | **25.1** |
| | dom_lu | **30.3** | **34.4** | **38.0** | **37.5** | **39.9** |
| *N. lasiophylla* | bio2 | **24.9** | **27.3** | **26.6** | **30.5** | **31.1** |
| | bio4 | 0.2 | 0.8 | 0.9 | 0.3 | 0.6 |
| | bio5 | 0.9 | 0.3 | 0.5 | 1.1 | 2.3 |
| | bio13 | 7.4 | 10.3 | 7.1 | 7.5 | 7.1 |
| | bio15 | **58.2** | **52.9** | **56.2** | **53.0** | **50.9** |
| | dom_lu | 8.4 | 8.3 | 8.6 | 7.7 | 8.0 |
| *N. lingulata* | bio2 | **19.4** | **20.6** | **21.6** | **23.0** | **23.0** |
| | bio4 | 9.8 | 4.9 | 4.1 | 4.9 | 3.3 |
| | bio5 | 4.9 | 3.8 | 3.1 | 1.8 | 1.2 |
| | bio13 | 1.8 | 6.9 | 6.2 | 6.0 | 5.7 |
| | bio15 | **43.0** | **41.0** | **42.5** | **40.5** | **42.2** |
| | dom_lu | **21.1** | **22.7** | **22.5** | **23.8** | **24.6** |
| *N. pulchella* | bio2 | **27.5** | **27.9** | **29.9** | **31.7** | **30.5** |
| | bio4 | **38.6** | **19.5** | **17.9** | **18.6** | **16.3** |
| | bio5 | 2.5 | 2.9 | 3.6 | 4.0 | 6.0 |
| | bio13 | 4.6 | 23.3 | 21.1 | 20.6 | 20.7 |
| | bio15 | 1.2 | 2.2 | 2.2 | 1.8 | 1.9 |
| | dom_lu | **25.6** | **24.4** | **25.2** | **23.4** | **24.6** |
| *N. rudgeana* | bio2 | **70.0** | **66.7** | **61.1** | **62.3** | **60.1** |
| | bio4 | 6.9 | 10.3 | 9.4 | 8.6 | 11.6 |
| | bio5 | 1.2 | 0.3 | 0.2 | 0.3 | 0.1 |
| | bio13 | 0.4 | 0.1 | 0.1 | 0.2 | 0.6 |
| | bio15 | 12.7 | 10.7 | 16.5 | 16.0 | 15.4 |
| | dom_lu | 8.8 | 11.8 | 12.7 | 12.6 | 12.2 |

### 3.3. Current Potential Distribution

The sampling of occurrence points indicated that *N. rudgeana* (12.90%), *N. amazonum* (10.75%), and *N. jamesoniana* (8.70%) had the highest percentage of points within the protected area, followed by *N. ampla* (3.13%), *N. lingulata* (2.13%), and *N. pulchella* (1.45%), while for *N. lasiophylla* all points were outside the protected area. The projected distribution maps for the seven *Nymphaea* species indicate Brazil and particularly eastern regions, Venezuela, Colombia, Ecuador, Bolivia, Guyana, and Peru (Figure 2a–g), to contain suitable habitat areas for the species. Some species had partial distribution, such as *N. ampla* in

Bolivia (Figure 2b), *N. pulchella* in Guyana (Figure 2f), and *N. rudgeana* in Bolivia and Peru (Figure 2g). Majority of the species were limited in distribution in states such as Argentina, Paraguay, Suriname, Chile, and Uruguay (Figure 2a–g). Generally, the eastern, northern parts, and some regions northwest of South America had the highest probabilities of suitable habitat, with certain regions in the central part of the continent also indicating potentially appropriate ranges. Among the seven water lily species, *N. jamesoniana* (2,790,903 km$^2$), *N. rudgeana* (2,657,233 km$^2$), and *N. amazonum* (2,339,884 km$^2$) occupied the most ranges in current distribution, followed by *N. ampla* (1,558,143 km$^2$), *N. lingulata* (1,341,940 km$^2$), *N. pulchella* (907,696.1 km$^2$), and *N. lasiophylla* (862,217 km$^2$), respectively (Table 2).

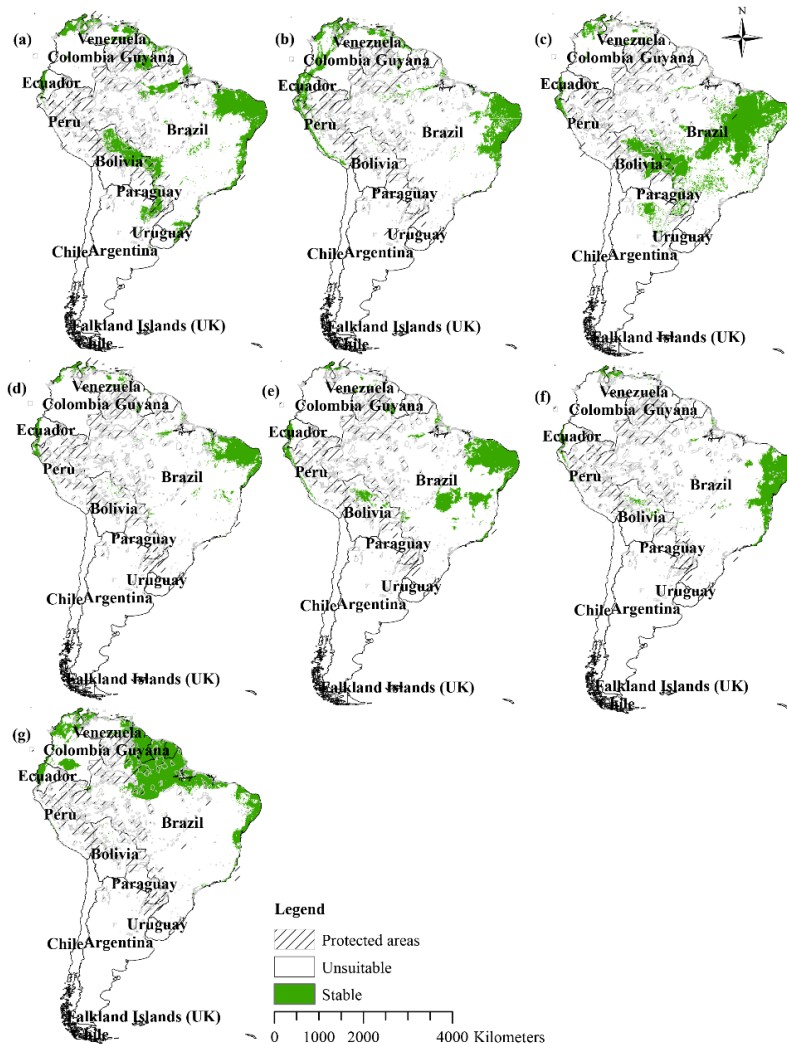

**Figure 2.** The current distribution of the seven *Nymphaea* species in South America: (**a**) *N. amazonum*, (**b**) *N. ampla*, (**c**) *N. jamesoniana*, (**d**) *N. lasiophylla*, (**e**) *N. lingulata*, (**f**) *N. pulchella*, and (**g**) *N. rudgeana*.

### 3.4. Future Distribution Changes

The assessment of the species' future distribution is important for conservation measures. The projection change summary for the distribution changes is presented in Figure 3 (Table S4). The future projection for *N. amazonum* indicates increased habitat gain and major habitat loss of suitability in Bolivia in RCP 4.5 (2050). Further loss of habitat suitability is projected consecutively in RCP 4.5 (2070) and both RCP 8.5 scenarios, especially in Bolivia and in most parts north of the continent, such as Venezuela, Guyana, and Suriname (Figure S1). The projection of *N. ampla* indicates a persistent habitat increase in RCP 4.5 (2070) and in both RCP 8.5 scenarios compared to RCP 4.5 (2050). Most habitat gain is projected in Brazil, Venezuela, and Guyana, while most parts of Peru and northwest of

the continent indicate habitat loss (Figure S2). Interestingly, *N. ampla* indicated increasing habitat expansion and loss in both RCP 4.5 scenarios and RCP 8.5 (2050), although habitat loss declined in RCP 8.5 (2070). The distribution of *N. jamesoniana* indicated greater changes in Brazil, Bolivia, Paraguay, and Argentina. These regions experience greater habitat loss compared to RCP 4.5 (2050), especially Brazil and Bolivia (Figure S3). However, in both RCP 8.5 scenarios, some regions in Brazil and Peru experienced an increase of suitable habitats, although habitat loss was increasing (Figures 3 and S3). The expansion ranges for *N. lasiophylla* in RCP 4.5 are lost in future scenarios of RCP 8.5, especially in Brazil and Guyana (Figure S4). *N. lasiophylla* indicated reduced habitat gain and increased habitat loss (Figure 3). *N. lingulata* was indicated to have the highest habitat gain in RCP 4.5 (2050) compared to all other species. Its expansion ranges are projected to contract in RCP 4.5 (2070) and both RCP 8.5 scenarios, especially in Brazil and Bolivia, which are characterized by continuous habitat loss (Figures 3 and S5). The projection of *N. pulchella* indicated greater habitat loss in Bolivia, Ecuador, and northeast of Brazil in both RCPs 4.5 and 8.5 (2050) (Figure S6). Although the expansion ranges are not obviously visible in the maps compared to the other species, Figure 3 indicates habitat gain in RCPs 4.5 and 8.5 of scenario 2070 compared to 2050. In scenario 2050 for both RCPs, *N. rudgeana* was projected to have a greater habitat increase compared to scenario 2070 for both RCPs (Figure 3). Many of the changes are noticed in central Brazil, Colombia, and Venezuela, with habitat expansion declining and habitat loss increasing (Figure S7). In extreme conditions of RCP 8.5 (2070), expansion ranges reduced greatly with increasing contractions (Figure 3).

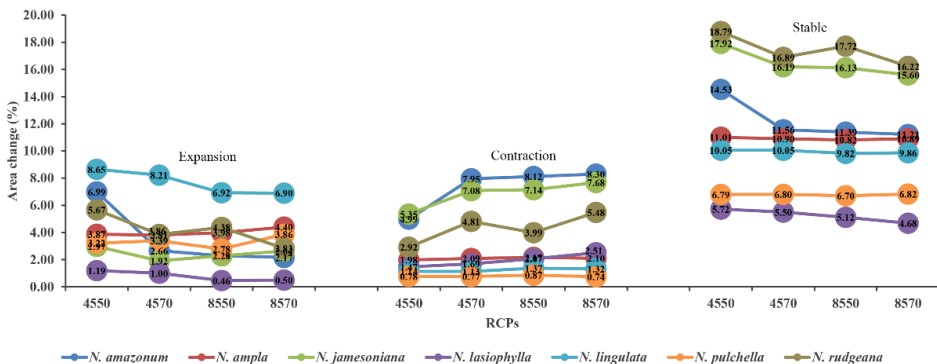

**Figure 3.** Future distribution changes in expansion (habitat gain), contraction (habitat loss), and stable environment among the seven *Nymphaea* species.

Generally, the species varied considerably in suitable habitat expansion compared to the contraction trend. Species such as *N. ampla*, *N. lasiophylla*, *N. lingulata*, and *N. pulchella* indicated an almost similar trend in habitat loss and the stable environment, although for *N. lasiophylla* the stable environment experienced shrinkage (Figure 3). The loss of suitable habitat expansion, stable environment, and the increase in contraction in the suitable ranges is a call for alarm when it comes to species' conservation. This loss is not only experienced by *N. lasiophylla* but also *N. amazonum* and *N. rudgeana* in RCP 8.5 (2070). *N. jamesoniana* indicates increased habitat expansions and contraction, and loss of stable environment, which indicate possible shifting to new areas. Lastly, many contractions occur in the most central parts of the continent compared to the coastlines, where many habitat growths and stable habitats are projected. These regions are therefore important in providing shifting ranges for suitable habitats.

### 3.5. Land Use and the Distribution of Water Lilies

The species-accessible area (M) for land use is projected in Figure 4. Most parts of the distribution regions remain unprotected, and this exposes them to human influence and degradation, more so the eastern parts of the continent (Table S5, Figure 4). Much influence is projected in the north, northwest, central, east, and southeast parts of the continent

according to the species-accessible area. Most of the areas are characterized by low land cover of between 50% and 75% (Figure 4). Land use was among the main contributing factors for the water lily species' distribution (Table 4). The current projection (Figure 2) and the future projections indicate that large areas of suitable habitats are projected outside PAs, areas acquired by accessible area classifications minus protected area classifications. The areas are characterized by grasslands or shrub areas, crop land areas, and areas composed of mixed activities (Figures S1–S7). Although large areas are covered by forests, especially in Brazil, crops, non-vegetation, and mixed activity areas are of significant influence to the continent (Table S5). Besides PAs playing a key role in the conservation and management of natural resources, they are limited in size compared to the un-PAs (Figure 5; Table S6) and therefore they can sometimes fail in ensuring sustainable refuge and survival for species under threat of extinction, especially when it is out of human control. In this case, the projection of the water lily species indicates an almost similar trend in expansion and contraction in both un-PAs and PAs, with the exception of *N. lingulata* which has higher expansion in un-PAs compared to PAs (Figure 5). Besides, Brazil, Ecuador, Colombia, Venezuela, and Bolivia are among the states with great habitat suitability and among those influenced by land use, more specifically Bolivia and Brazil, where many of the projected areas are declining. Besides, the stable environment had no significant change between un-PAs and PAs (Figure 6). The un-PAs had much of the stable area among the species, except for *N. rudgeana,* for which in both areas the stable environment was almost the same (Figure 6).

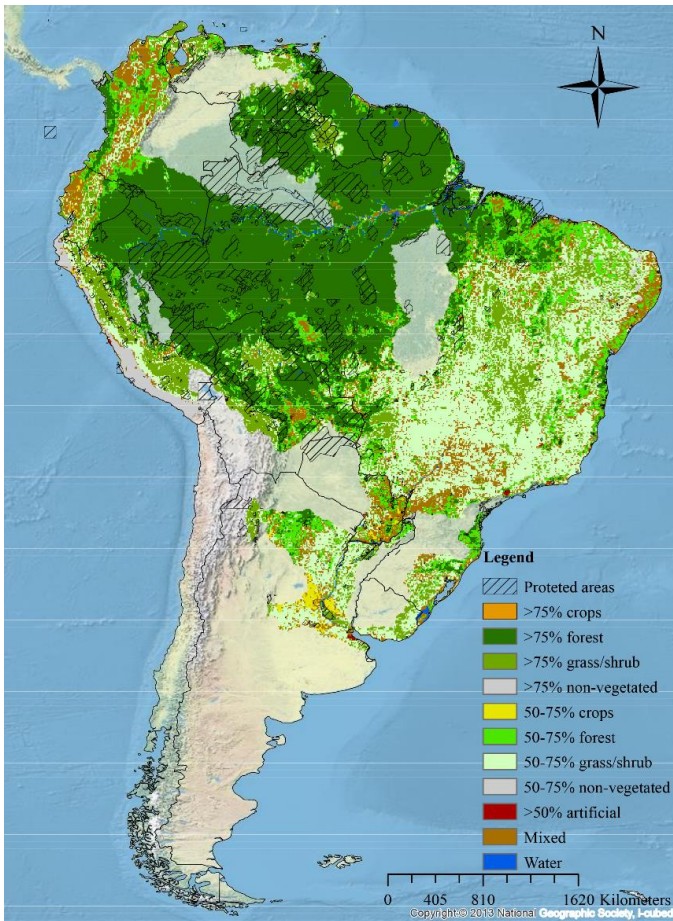

**Figure 4.** Land use characteristics in South America within the water lily (*Nymphaea*)-accessible area (M), indicating the land use category in protected and unprotected areas.

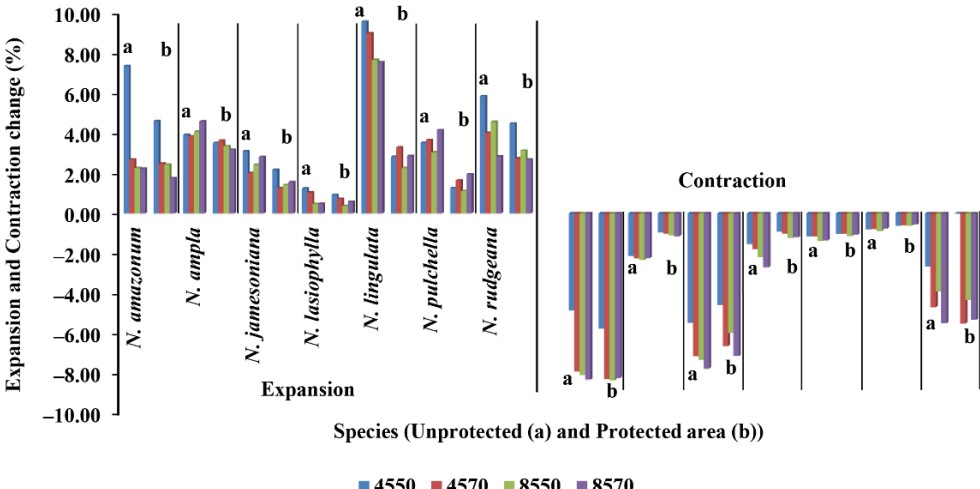

**Figure 5.** Comparison of habitat suitability change for the water lily species within unprotected areas (a) and protected areas (b). Change area is represented in percentage, expansion in positive values, and contraction in negative values.

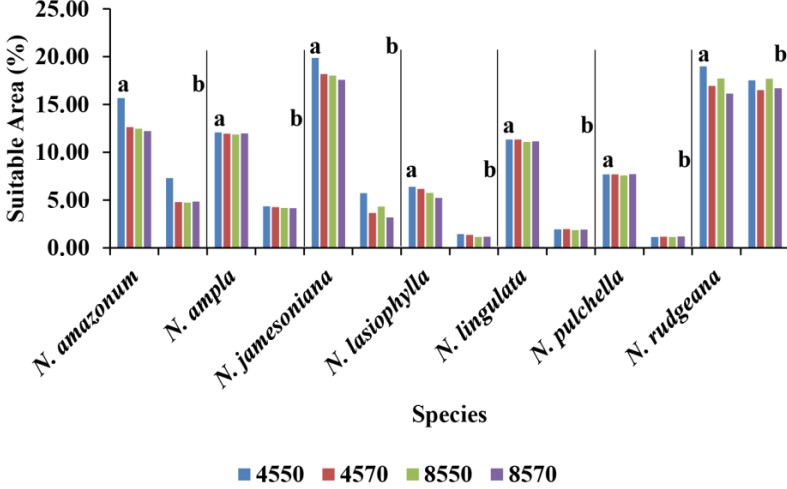

**Figure 6.** Comparison of the stable habitat change for the water lily species within unprotected areas (a) and protected areas (b). Change area is represented in percentage.

## 4. Discussion

This study developed a set of potential ecological niche models for the *Nymphaea* species to examine their distribution patterns across South America. We used species occurrence data and bioclimatic variables to simulate species' ecological niches. These factors have been proven fundamental for MaxEnt modeling [38]. As a result of the variable selection procedure, the VIF values were less than three, indicating less collinearity [42]. The rm values in the selected feature classes were higher compared to the MaxEnt default rm of one, thus highly minimizing the overfitting of the models [43]. The mean AUC values obtained from the MaxEnt models reflected the common values expected for fairly fitted models (Table 2) and were not significantly different from previous studies, and were thus considered descriptive [16,24,44].

In this study, temperature and precipitation variables (bio2, bio13, and bio15) influenced *Nymphaea* species' distribution in South America similarly to the African *Nymphaea* species [16]. For instance, bio2 was predicted as the primary temperature variable with the greatest impact in all the studied *Nymphaea* species except *N. jamesoniana* in South America. The mean diurnal range (bio2) indicates the variation in temperature over a day, and the

temperature changes may have a significant impact on plant growth, especially photosynthesis and respiration of the plant, contributing to nutrient buildup [45]. Bio4 and bio15 mediate species' suitable distribution by forcing them to shift between favorable seasons. In this case, the species are likely to shift to regions experiencing less dry seasons and cooler temperatures [46]. Unlike bio13, which provides a more conducive environment by increasing precipitation that replenishes the isolated habitats such as ponds and increasing surface runoff, bio5 increases evaporation and transpiration, which significantly affect the plants and their environment by causing dryness. As a result, the plants are forced to seek suitable locations. Lastly, land use directly declines species' habitats depending on the magnitude of land use.

The current projected distribution of the *Nymphaea* species revealed areas of high environmental suitability that corresponded to the observed records. This might indicate that ecological niches are defined by variables affecting the distribution and abundance of resources on which they rely on [47]. Nevertheless, due to continuous habitat destruction on the continent, the projected suitable habitat areas may over-represent habitat suitability where *Nymphaea* species habitats no longer exist or their habitats are under the extreme pressure of sustaining them, such as in the northeast of Brazil where human influence is high [8]. Greater changes in habitat suitability were noticed in the future distribution of *N. amazonum*, *N. jamesoniana*, and *N. rudgeana*, which may have been caused by the reduced moisture, increasing aridity in multiple locations, and precipitation variability [48]. Such situations are propagated by temperature increases, especially in the warm months, which increase evaporation and dryness; for example, the contribution of bio5 for *N. amazonum* and *N. jamesoniana* distribution. As global temperatures rise, the threat of heat stress increases, possibly causing a decrease in species-suitable habitats and richness [3,49]. The rise in temperature may push the species to adjust in habitat-suitable areas, thus causing geographical shifts [50].

Although the environmental requirements for the *Nymphaea* species slightly differed, these species utilized similar habitats in the northeastern parts of the continent. Moreover, it is proposed that in most wetland ecosystems, aquatic species are sympatric [11]. Similarly, in the South American freshwater ecoregions, such relationships between *Nymphaea* species have been documented [47,51]. Furthermore, *Nymphaea* species would display niche partitioning and geographic segregation in South America [52].

The protected areas (PAs) act as shelter for species' survival in this human-dominated world [53]. Species take refuge in the uninfluenced ecosystems, thus ensuring their survival. Although the regions hold biodiversity and conservation value, they are also under extreme pressure from the growing human populations and need for development resources [54]. Nonetheless, with the high magnitude of global warming, protected areas also face adversities in conservation, resulting in declining habitat suitability and vegetation shifts [55]. In this study, a similar change in habitat suitability was observed between PAs and un-PAs in habitat suitability. This could imply a declining capability of PAs to provide a safe refuge for species with declining habitats [56,57]. The only hope with this kind of projection is that the declining habitats in PAs will happen slower and more visibly compared to the non-PAs due to regulated human impacts. However, it is a wakeup call for conservation managers to rethink on the impact of PAs on some species as climate change will cut across all environments.

Surprisingly, the Amazon ecosystem was thought to provide the most suitable habitat for the *Nymphaea* species, however the species' limited distribution in the area was linked to the increasing climatic events such as drought and numerous floods which destroy the species' habitat [2]. Besides, the Amazon's humid environment may not be suitable for numerous growth of the water lily species compared to the savannah areas, where the habitat has been favored [15]. Since habitat suitability distribution in the Amazon might be controlled by many factors, we leave it as an open gap for further studies involving field surveys which can provide the absent data that are capable of improving our model in confirming the water lily species' distribution in the region. Unlike the PAs, much

distribution is projected in human-disturbed environments, especially in some parts of the Amazon, which could support the hypothesis that deforestation reduces evapotranspiration and increases stream flow [58]. In addition, human influence through land use changes may have a significant impact on land water resources by affecting the mechanisms at which latent heat flux, surface runoff, discharge from the rivers, and regional and continental precipitation patterns occur [59].

In conclusion, this study provided insight on the implications of climate change and land use effects in assessing the potential habitat suitability for the *Nymphaea* species as the main factors influencing aquatic ecosystems and biodiversity. The study is important for the conservation and management of these species, which are scientifically and ecologically important. The loss of suitable habitat areas always raises concern for possible changes, leading to the reduction of species' populations that may result in species being threatened or becoming extinct [60]. As PAs face increased protection and conservation crises, it is high time for the conservationists to turn to and engage the local indigenous people, civil societies, and private sectors in biodiversity conservation and management [61]. Such measures have been implemented in states such as Bolivia, Peru, Guyana, and other areas outside South America, and they may be of higher significance in the conservation of the South American wetland ecosystem and biodiversity [62].

**Supplementary Materials:** The following supporting information can be downloaded at: https://www.mdpi.com/article/10.3390/d14100830/s1, Figures S1–S7: The projected distribution change for the seven *Nymphaea* species for the future projection scenarios. (Figure S1) *N. amazonum*, (Figure S2) *N. ampla*, (Figure S3) *N. jamesoniana*, (Figure S4) *N. lasiophylla*, (Figure S5) *N. lingulata*, (Figure S6) *N. pulchella*, and (Figure S7) *N. rudgeana*. Table S1: The occurrence points for the seven water lily (*Nymphaea*) species used for this study. Table S2: The sixteen variables used for correlation in variance inflation factor (VIF). Table S3: Training (75%) and testing (25%) AUC values and their standard deviation values for the *Nymphaea* species' distribution suitability using the maximum training sensitivity plus specificity logistic threshold (MTSS). The AUC values describe the fitness of the model in predicting the species' distribution. Table S4: Distribution change obtained by comparing binary changes between the current and future potential distribution for the seven water lily species (*Nymphaea*) in South America. Table S5: Projected area size between protected areas and unprotected areas for the seven *Nymphaea* species' accessible areas in South America. Table S6: Distribution change obtained by comparing binary changes between the current and future potential distribution for the seven water lily species (*Nymphaea*) in protected areas of South America.

**Author Contributions:** J.M.N.: conceptualization, investigation, formal analysis, writing—original draft; B.K.N.: formal analysis, writing—review and editing; V.M.M.: conceptualization, writing—review and editing; J.K.K.: writing—review and editing; Q.-F.W.: resources, funding acquisition; C.P. and J.-M.C.: conceptualization, resources, supervision; Z.-Z.L.: supervision, writing—review and editing. All authors have read and agreed to the published version of the manuscript.

**Funding:** This work was supported by grants from the National Natural Science Foundation of China, China (No. 32070231), and CAS-TWAS President's Ph.D. Fellowship Program University of the Chinese Academy of Sciences, China.

**Institutional Review Board Statement:** Not applicable.

**Data Availability Statement:** All data presented in this study are available in the article and in the Supplementary Materials.

**Conflicts of Interest:** The authors declare no conflict of interest.

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
