# Peer review of "Assessment of Climate Change and Land Use Effects on Water Lily (Nymphaea L.) Habitat Suitability in South America"

_diversity, doi:10.3390/d14100830_

Round 1

Reviewer 1 Report

In the manuscript »Assessment of climate change and land use effects on water lily (Nymphaea L.) habitat suitability in South America «, authors John M. Nzei, Boniface K. Ngarega, Virginia M. Mwanzia, Joseph K. Kurauka, Qing-Feng Wang,  Jin-Ming Chen, Zhi-Zhong Li , and Cheng Pan, assessed the habitat distribution for seven Nymphaea species in South America.

Abstract

The following statement is unclear:

L 32-34  For example, N. amazonum, N. rudgeana, and N. lasiophylla future expansions decline with increasing contraction, N. ampla and  N. jamesoniana suitable habitats expanded with an increase in contractions while N. pulchella and N. rudgeana habitat suitability is variable in the RCPs.

OK.

Key words Are OK.

Introduction

L 72-74. The following statement is not clear: Most species are sampled in Brazil and  less in other parts of the continent, which might signify variability in the availability of  the species although sometimes sampling is biased to the most accessible areas.

L 82 Use superscript (560,000 Km2) of…

Clear with the relevant references cited.

Materials and Methods

Methods are properly described.

Results

Legend is missing after the tables 1, 2, 3 and 4

L 180 Table2)… space is missing

L 221 – 223. Where did you get the folowing data? Among the seven waterlily species, N. jamesoniana (2,790,903 Km2), N. rudgeana (2,657,233 Km2), and N. amazonum (2,339,884 Km2) occupied the most ranges, followed by N. ampla (1,558,143 Km2), N. lingulata (1,341,940 Km2), N. pulchella (907,696.1 223 Km2), and N. lasiophylla (862,217 Km2) respectively (Table 2).

L 232 figure 3 Capital letter for Figure

L 269: N. jamesoniana  indicate increased… indicated or indicates?

L 278 north, North West, central parts, east, and south east parts… Capital letters?

Discussion

L 392 extinct[59]…. Space is missing

Specific comments

Is the ecological requirements for Nymphaea species in the study area available?

The authors present the study of practical relevance. The study was well planned and performed and it collects a series of measures. The manuscript presents new findings.

My suggestions: minor revision

Author Response

Our response to reviewer 1 is attached as a file

Reviewer 2 Report

The presented manuscript focuses on changes in the potential range of seven species of the genus Nymphaea in South America. Standard methods were used, although the number of replications (10) is relatively small. The results obtained seem consistent. However, the paper contains some errors and shortcomings. The English language has room for improvement, especially in the introduction, where there is a lot of unclear wording. In the discussion, the authors note that the results obtained do not predict the widespread occurrence of the species in the seemingly good conditions of the Amazon forest. Unfortunately, they do not discuss this in much detail - and this result may be related to the nature of the input data. The analyses were performed using locations included in the GBIF database. Because of this, observations are statistically more frequent in more densely populated areas (such as the northeastern coast of Brazil), while they are lacking in inaccessible locations such as the Amazon forest. This potential bias in the results obtained should be discussed. The authors should also simplify the description of the results, as writing about simultaneous expansion and contraction can cause confusion. I understand that the authors meant gain and loss of suitability in different areas of species range, which leads to a shift of this range. However, this could have been worded more clearly. The work can be regarded as a rough estimate of the potential range of the studied species, but the manuscript still needs work to better describe and interpret the results obtained.

Other comments:

L31 What means „variable expansion”?

L32-35 This sentence is not clear; “habitats expanded with an increase in contractions” seems to be contradictory

L64 What means “ecosystem manipulation”?

L70 “poorly unexplored and undocumented” should be “poorly explored and documented”

L71-74 This part is unclear

L110 For each species it should specific link to the GBIF data, as it is described in GBIF guideline (https://www.gbif.org/citation-guidelines)

L133 “Km” should be “km” (it should be checked along the text)

L136 The authors removed four of presence of artefacts; however, these variables present strict border between pixels, because warmest/coldest and wettest/driest quarters consist of different months in different areas, thus, the borders looks artificially. I suggest to check, whether these variables are correlated with used factors; for water species, the variable that show temperature-precipitation connection may be significant.

L178 It should be simply “VIF values”

L258, Fig 3. It is not clear how the “stable” areas are connected with “expansion” and “contraction”. If some part of range start to be unsuitable, it is treat as a drop in the “stable” class or only “contraction”?

L278 Why “North West” is wrote by capitals, whereas all others are by small letter?

Author Response

Our response to reviewer 2 has been attached as file

Round 2

Reviewer 2 Report

Manucript was corrected.